# Constipation-Predominant Irritable Bowel Syndrome (IBS-C): Effects of Different Nutritional Patterns on Intestinal Dysbiosis and Symptoms

**DOI:** 10.3390/nu15071647

**Published:** 2023-03-28

**Authors:** Claudia Di Rosa, Annamaria Altomare, Vittoria Terrigno, Florencia Carbone, Jan Tack, Michele Cicala, Michele Pier Luca Guarino

**Affiliations:** 1Research Unit of Food Science and Human Nutrition, Department of Science and Technology for Humans and the Environment, Università Campus Bio-Medico di Roma, Via Alvaro del Portillo, 21-00128 Roma, Italy; c.dirosa@unicampus.it (C.D.R.); terrigno.v@gmail.com (V.T.); 2Research Unit of Gastroenterology, Università Campus Bio-Medico di Roma, Via Alvaro del Portillo, 21-00128 Roma, Italy; m.cicala@policlinicocampus.it (M.C.); m.guarino@policlinicocampus.it (M.P.L.G.); 3Operative Research Unit of Gastroenterology, Fondazione Policlinico Universitario Campus Bio-Medico, Via Alvaro del Portillo, 200-00128 Roma, Italy; 4Department of Gastroenterology and Hepatology, University Hospitals Leuven, 3000 Leuven, Belgium; florencia.carbone@kuleuven.be (F.C.); jan.tack@kuleuven.be (J.T.)

**Keywords:** irritable bowel syndrome and diet, irritable bowel syndrome with constipation, IBS and diet, IBS and microbiota

## Abstract

Irritable bowel syndrome (IBS) is a chronic functional gastrointestinal disorder characterized by abdominal pain associated with defecation or a change in bowel habits. The pathogenesis of IBS is not completely clear, but it is known to be multifactorial and complex. Endogenous and exogenous factors such as abnormal GI motility, low-grade inflammation, increased epithelial permeability and visceral hypersensitivity, but diet and psychosocial aspects are also recognized as important actors. Furthermore, the interaction between diet and gut microbiota has gained interest as a potential contributor to the pathophysiology of IBS. To date, there is no specific diet for IBS with constipation (IBS-C); however, many studies show that fiber intake, especially soluble fiber such as inulin, could have a positive effect on symptoms. This review aims to evaluate the effects of some nutritional components such as fibers but also functional foods, prebiotics, probiotics and symbiotics on symptoms and microbiota in IBS-C subjects.

## 1. Introduction

Irritable bowel syndrome (IBS) is a functional gastrointestinal disorder characterized by recurrent abdominal pain and discomfort associated with altered bowel habits that occur in the absence of structural and biochemical alterations or other organic gastrointestinal (GI) diseases [1,2,3].

The main symptoms of IBS are abdominal pain, cramps, constipation or diarrhea, bloating and changes in stool pattern. Based on the Rome IV criteria, the diagnosis of IBS includes recurrent abdominal pain, present at least 1 day/week in the last 3 months, with at least two of the following criteria: (1) related to defecation; (2) associated with a change in the frequency of stools; (3) associated with a change in the form of stools. These criteria must have been present for the last 3 months, and the symptoms must have started at least 6 months before diagnosis [4].

Four different subtypes of IBS are identified according to the shape and consistence of stools, referring to the Bristol Stool Scale (BSS) [4]:

Constipation-predominant IBS (IBS-C): more than 25% of bowel movements are classified as BSS 1 or 2, and less than 25% as 6 or 7.Diarrhea-predominant IBS (IBS-D): more than 25% of stools are categorized as BSS 6 or 7, and less than 25% as 1 or 2.Mixed bowel habits IBS (IBS-M): more than 25% are constipated (type 1 or 2) and more than 25% diarrhea (type 6 or 7) stools.Unclassified IBS (IBS-U): the symptoms meet other criteria for IBS, but no more than 25% of stools is abnormal.

The worldwide prevalence of IBS is around 4% [5], making it one of the most diagnosed GI disorders [5] with a higher prevalence (2:1 ratio) in women [5,6]. IBS is more present in childhood, even if it seems to have the peak of prevalence in the early adulthood [2,7]. Around 30% of people affected by IBS consult physicians, not always for severe abdominal symptoms, but especially because they have high levels of anxiety and a low quality of life (QoL). IBS patients generally present also other functional diseases and usually undergo more surgeries than the general population, possibly erroneously aiming at improving the abdominal symptoms [6]. Although there is no excess of mortality associated with IBS [8], this disorder can considerably impair the quality of life [9].

Despite its high prevalence, the pathophysiology of IBS is not completely known yet, and seems to be heterogeneous and multifactorial [10,11]. Indeed, several factors are involved in its pathophysiology, such as alterations in gastrointestinal motility, visceral hypersensitivity, small intestinal bacterial overgrowth (SIBO), environmental factors, dietary habits, food intolerances [12] and intestinal microbiota alterations (dysbiosis) [13]. To date, it is unclear if dysbiosis is a cause or a consequence in the pathogenesis of this syndrome. However, there is a large amount of evidence supporting the involvement of the gut microbiota in the pathophysiology of IBS, and both qualitative and quantitative alterations in gut microbiota have been observed [14,15]. It is well known that microbial changes could worsen gut symptoms associated with IBS, such as visceral pain, low-grade inflammation, and changes in bowel habits [16].

It has also been convincingly demonstrated that diet can affect IBS, and this is the reason why many people tend to associate the various symptoms with nutrition. As a result, many patients tend to exclude certain foods from their diet on their own, or follow unsuitable nutritional patterns [17,18]. Although in the literature there is a lack of trials that evaluate the role of diet in the IBS-C subtype, this review aims to focus especially on the effects of fiber consumption on symptoms and microbiota.

## 2. Search Strategy

Literature research was conducted to identify clinical trials investigating the effects of diet on irritable bowel syndrome with constipation. We searched for papers published up to December 2022 to evaluate the most recent findings in this field across the PubMed database, with the following keywords: “irritable bowel syndrome and diet”, “irritable bowel syndrome and constipation”, “IBS and diet”, “IBS and microbiota”. The inclusion criteria identified original articles and reviews published in English on an adult population with IBS, especially IBS-C. Unpublished studies or studies published in languages other than English were excluded. Eligible studies were firstly evaluated ono the basis of the abstract and then included in the manuscript if they met the inclusion criteria. An additional manual search was conducted through citations of included articles to identify the most suitable works.

## 3. Human Gut Microbiota: A Brief Description

The GI tract houses 10^14^ bacteria and it comprises 150 times more genes than the human genome. The microbiota is a metabolically active community that exerts important influences in health and disease, and the relationship between the host and its gut microbiota has been described as a mutualistic ecosystem in which both benefit [19]. The most dominant bacterial phyla in the human gut are Firmicutes, Bacteroidetes, Actinobacteria, and Proteobacteria, and the most recorded bacterial genera are *Bacteroides, Clostridium, Peptococcus, Bifidobacterium, Eubacterium, Ruminococcus, Faecalibacterium* and *Peptostreptococcus*. The human microbiota is established after birth, and it is dominated by *Bifidobacterium* (Actinobacteria) [20]; however, during growth, the microbial composition changes both in diversity and richness [21], reaching the highest complexity, with several hundred species-level phylotypes dominated by Bacteroidetes and Firmicutes [22].

The gut microbiota is involved in several important functions in the human body:(1)The production of different antimicrobial substances to defend the host, thereby enhancing the immune system [23];(2)The digestion and metabolism of dietary components [24];(3)The proliferation and differentiation control of epithelial cells [25];(4)The gut–brain communication influencing the host’s mental and neurological functions [26];(5)The maintenance of the normal gut physiology and health [27];(6)The fecal mass production decreasing the transit time and diluting the toxic substances that affect the health of the host [28].

Two distinct GI microbiota populations are recognized: one within the colonic lumen and the other one adherent to the epithelium mucosa [29]. The first one, very variable, is measurable via stool sampling and is a combination of non-adherent luminal bacteria with a mix of shed mucosal bacteria [30]. On the contrary, the mucosal population is stable in an individual [31], and is involved in the ‘crosstalk’ between the lumen and the tissue under the mucosal border at which the interaction between immune and enteroendocrine cells occurs [32].

The microbiota community may change according to host factors (gender, age, body weight, diet, drug exposure, pathological conditions) and is determined by the adaptability of the organism’s phenotype, the physical environmental conditions of the gastrointestinal tract (e.g., gastric acid, gastrointestinal motility and secretions), genetic factors and colonization. Both the pH and the oxygen availability affect the spatial distribution of the microbiota populations [33]. In the upper GI tract, both lower pH and fast transit inhibit the bacterial growth, while the bacterial density and diversity increase gradually from the stomach to the colon (10^12^ cells/mL) [30], in which mainly anaerobic microorganisms such as Bacteroidetes*, Porphyromonas, Bifidobacterium, Lactobacillus and Clostridium* live with a ratio anaerobe:aerobes of 100-1.000:1, thanks to the very low concentration of oxygen [34].

Because of the complex interaction between diet and the gut, the microbiota within the gut lumen contributes to the production of short chain fatty acids (SCAFs): butyrate, propionate and acetate. These are the primary end-products of fermentation of non-digestible carbohydrates (NDC) that are used as nourishment for the gut microbiota [35]. The acetate production pathways are common among bacterial groups, whereas pathways for propionate and butyrate production are more substrate-specific. Species such as *Akkermansia municiphila* have been identified as key propionate-producing and mucin-degrading microorganisms, and interestingly, an increase in their phylogenetic variability in IBS patients has been identified [36]. On the other hand, a surprisingly small number of microorganisms, such as *Faecalibacterium prausnitzii, Eubacterium rectale, Eubacterium hallii* and *R. bromii*, appear to be responsible for butyrate production [35].

The healthy composition of the gut microbiota is the main prerequisite for the proper functioning of the other organs. When the mutualistic relationship among microbiota members and the host is lost, a condition called dysbiosis occurs. In this case, potentially pathogenic microbes, called pathobionts, prevail at the expense of potentially beneficial ones [37,38], determining the possible development of pathological conditions in the whole organism.

## 4. Gut Microbiota in Irritable Bowel Syndrome with Constipation

Various factors are involved in the pathogenesis of IBS-C, such as the type of diet, genetic predisposition, colonic motility, absorption, social-economic status, daily behaviors, and biological and pharmaceutical factors. Furthermore, low fiber dietary intake, inadequate water intake, sedentary lifestyle and failure to respond to the urgency to defecate have been revealed to constitute a predisposition [39].

In the different IBS subtypes, diverse microbial populations have been observed in the microbiota adherent to the mucosa. Through bacterial culture tests, Malinen et al., showed that IBS-C patients had significantly increased levels of *Veillonella* species compared to healthy controls (*p* < 0.045), as well as higher levels of *Lactobacillum* compared to IBS-D patients [40]. Interestingly, Maukonen et al., through analyzing fecal DNA revealed that 30% of all bacterial species in IBS-C patients were *Clostridium coccoides* and *Eubacterium rectale*, which was significantly lower compared to healthy control subjects (43%) and IBS-D patients (50%) [41]. Rajilić-Stojanović et al., demonstrated that IBS-C patients presented a significantly higher level of Firmicutes, including *Clostridium* species (*p* < 0.05), and a significantly lower level of Actinobacteria and Bacteroidetes (*p* < 0.01) than healthy controls [42]. In a fecal culture experiment, Chassard et al., observed that IBS-C patients had significantly higher levels of Enterobacteriaceae (*p* = 0.0107) and sulfate-reducing bacteria, and significantly lower levels of *Bifidobacterium* (*p* < 0.0001) and *Lactobacillus* (*p* = 0.0007) than healthy control subjects [43]. In studies examining mucosal bacteria through 16S rRNA metagenomic analysis, Durbán et al., found an increased level of Bacteroidetes and Enterobacteriaceae in IBS-C patients [44]; in addition, Parkes et al., demonstrated that IBS-C patients had a higher level of Bacteroidetes, *Bifidobacterium*, and *C. coccoides/E. rectale* [45] (Table 1). Moreover, even if a clear consensus does not exist, many studies suggest that IBS-C patients, compared to healthy subjects, have a lower presence of Actinobacteria and a higher level of Bacteroidetes in their fecal samples.

A recent study identified the different micro types linked to IBS-C and IBS-D subtypes, and found that IBS-C subjects had a higher breath methane production directly proportional to the higher relative abundance of Methanogens [46], especially *Methanobrevibacter smithii,* that was observed in stool [46,47].

IBS-C patients with noticeable methane production showed higher constipation severity scores, and the amount of methane detected in the breath test was directly proportionate to the degree of reported constipation [47]. In fact, methane decreases ileal and colonic transit time and decreases the amplitude of contraction, slowing peristalsis and causing constipation. Studies in the literature have shown that affecting methanogenesis directly causes improvement of IBS-C symptoms [48].

Moreover, in IBS-C patients, alongside methane producers, a higher relative abundance of hydrogen producers such as Ruminococcaceae and Christensenellaceae, has been observed, and their amount was directly correlated to Methanogens’ abundance [46].

Another study comparing the microbiota of IBS-C patients and healthy controls found that there were no significant differences in fermentative patterns of methane and nonmethane production in fecal samples in both groups, except for starch fermentation, which determined a higher gaseous metabolite production in IBS-C patients. Additionally, regarding acetate and propionate production, no significant differences were found between IBS-C and healthy subjects [43]. Therefore, there is convincing evidence that the microbiota is a predominant factor in IBS-C pathophysiology, and that its modulation may be part of future therapeutic approaches to managing IBS.

## 5. Dietary Approaches Suggested in Patients with IBS-C

### 5.1. Traditional Dietary Advice

General advice on healthy eating and lifestyle is recommended as the first-line intervention in the dietary management of IBS [49]. Dietary advice is based on the National Institute for Health and Care Excellence (NICE) [50] and the British Dietetic Association [51] guidelines. People with IBS should be warned about the importance of self-management, including information on general lifestyle, physical activity, diet and symptom-targeted medication [50]. Healthcare professionals should assess the physical activity levels of people with IBS, for example, using the General Practice Physical Activity Questionnaire (GPPAQ). People with low activity levels should be encouraged to increase their physical activity levels [50] and they should be encouraged to follow the following general advice [50,51]:Eat regular meals, taking time to eat.Avoid skipping meals or leaving long time-spans between meals.Drink at least 8 cups of water or non-caffeinated drinks per day, reducing the intake of alcohol and fizzy drinks.Restrict tea and coffee to 3 cups per day.Reduce intake of ‘resistant starch’ (starch that resists the digestion in the small intestine and reaches the colon intact), which is often found in processed or pre-cooked foods.Limit fresh fruit intake to 3 portions per day (a portion should be approximately 80 g).People with diarrhea should avoid sorbitol, an artificial sweetener found in sugar-free sweets (including chewing gum) and drinks, and in some diabetic and slimming products.

### 5.2. FODMAP-Restricted Diet and Fiber in the Management of IBS-C

In general, a low FODMAP diet has been being proposed for several years for the nutritional management of IBS. FODMAPs are a group of fermentable carbohydrates (Fermentable Oligosaccharides, Disaccharides, Monosaccharides, Additionally, Polyols), which include fructo-oligosaccharides (FOS), galacto-oligosaccharides (GOS), disaccharides (e.g., lactose), monosaccharides (e.g., fructose) and polyols (e.g., sorbitol). The dietary restriction of FODMAPs is now increasingly used in the clinical setting. A large number of individuals are sensitive to FODMAPs, which can be associated with increased symptoms such as bloating, diarrhea, gas, constipation, or abdominal pain that are reminiscent of IBS symptoms. FODMAPs if malabsorbed exert a highly osmotic effect that can cause an influx of water into the colon, resulting in diarrhea, or they may be fermented by colonic bacteria, leading to excessive gas production. The visceral hypersensitivity usually present in IBS subjects may worsen after the intestinal distension triggered by gas or fluids, thereby leading to IBS abdominal symptoms [37]. Theoretically, reduced consumption of FODMAPs would reduce fluid accumulation in the gut and improve symptoms [52]. Nevertheless, a low-FODMAPs diet also decreases fiber intake, and thus might cause constipation in some patients [52].

Interestingly, Nybacka et al., aimed to study the relationship between habitual FODMAP intake and symptom severity. One hundred and eighty-nine patients with IBS, of which 54 had IBS-D (27.4%), 46 had IBS m (23.4%), 46 had IBS-U (23.4%) and 44 had IBS-C (22.3%), were enrolled, and they recorded their food intake for four days. Symptom severity was measured with the IBS severity scoring system (IBS-SSS) [53], which is a validated questionnaire to evaluate the severity of IBS symptoms. The study demonstrated small differences in FODMAPs intake among different IBS subtypes. Results showed that the FODMAPs consumption exerts different effects in individuals with IBS, depending on the subtype. As the different FODMAPs seem to be more or less involved in generating symptoms, it is necessary to evaluate the effect of each FODMAP separately in each IBS subtype, in RCTs or longitudinal studies [53].

In 2010, Ong et al., in their single-blind cross-over study recruited 15 healthy subjects and 15 subjects with IBS to investigate the FODMAP-restricted diet in Australia. Out of the 15 IBS subjects, seven had IBS-C, four had IBS D, two had IBS m and two patients had IBS-U. Participants followed a FODMAP-restricted diet (9 g/day) or a high-FODMAP diet (50 g/day) for 2 days each with a 7-day wash-out period between the two diets. Abdominal pain (*p* = 0.006), bloating (*p* = 0.002), the passage of gas (*p* = 0.002) and nausea (*p* = 0.01) significantly reduced during the FODMAP-restricted diet. The passage of gas was also significantly lower in healthy subjects while they were on the FODMAP-restricted diet (*p* = 0.007). Other symptoms did not change during the two different diets for healthy individuals [54].

Tuck et al., conducted a randomized, double-blind, placebo-controlled, cross-over trial to evaluate whether oral α-galactosidase co-ingestion with foods high in GOS and low in other FODMAPs would reduce symptoms. Thirty-one patients with IBS (20 IBS-D, 7 IBS-M, 4 IBS-C) completed the study, and 21 out of 31 participants were identified as “GOS-sensitive”. All subjects in the IBS m subgroup presented GOS sensitivity, with worsening overall symptoms during the high-GOS diet compared to the low-FODMAP diet, while there were no statistical differences in the IBS-C participants [55]. However, the co-ingestion of the α-galactosidase significantly reduced the GOS-induced symptoms.

A recently published study, the DOMINO trial, showed that a FODMAP-lowering diet was more effective than standard medical therapy in alleviating IBS symptoms in primary care. Four hundred and fifty-nine newly diagnosed IBS subjects were randomized to the diet arm (*n* = 227) and the medication arm (*n* = 232) for 8 weeks and underwent two follow-up visits (after 16 and 24 weeks). Dietary advice was in accordance with NICE guidelines and aimed to induce a mild reduction in FODMAPs; it was administered via an electronic application that allowed patients to be more independent. The diet group showed a significantly greater improvement in symptoms (IBS-SSS) compared to otilonium bromide (OB), a musculotropic spasmolytic. The dietary application was associated with a significantly higher response rate, with an improvement in IBS-SSS [56].

Overall, what emerged from these studies is that a low-FODMAP diet leads to an improvement in IBS symptoms; however, further studies are needed to demonstrate statistical significance, especially in patients with IBS-C.

On the other hand, dietary fiber supplements have been advocated for the management of IBS-C.

Among the fibers, soluble ones should be recommended, such as oat, *Psyllium* (Ispaghula) or inulin; in fact, this kind of fiber may have potential beneficial effects on the QoL and bowel function in IBS-C patients in terms of stool frequency, consistency and transit time [57]. On the other hand, unfortunately, consuming dietary fibers also determines an increase in the osmotic effect, and consequently, an increase in the colonic content volume with distension of the abdominal walls, which may also be responsible for symptom generation. That said, this problem appears to be mainly related to consumption of insoluble fibers rather than soluble ones.

In IBS, insoluble fiber may exacerbate symptoms and provide minimal relief, but soluble fiber such as psyllium, can be effective in reducing symptoms. To date, there is recent evidence for fiber supplementation and for a FODMAP-restricted diet in the management of IBS-C. Saulnier et al., evaluated gut microbiota changes and pain response in children with IBS after a low- and a high-FODMAP diet for 2 days. Patients were categorized as responders, non-responders, and placebo-responders. An abundance of baseline taxa known as carbohydrate fermenters (*Bacteroides, Ruminoccus, Faecalibacterium prausnitzii*) was found in the responder group. It has been concluded that IBS patients with saccharolytic enriched microbiota may benefit most from a low-FODMAP diet. It is important to only limit the FODMAPs that can exacerbate symptoms, and not all FODMAPs, as this second case could have negative long-term outcomes on the microbiome and on the intestinal integrity. This was highlighted in a study that found that a low-FODMAP diet, while improving IBS symptoms, also depleted levels of butyrate-producing bacteria from *Clostridium* cluster XIVa and mucus-associated bacteria [58]. In conclusion, recent studies showed that dietary interventions with natural fiber or fiber supplements can be useful for the management of patients with IBS-C, and likewise FODMAPs [58].

### 5.3. Soluble and Insoluble Fibers in IBS-C

Diet plays a pivotal role both in the pathophysiology of IBS, and in the improvement of symptoms and QoL [34].

Stephen and Cummings [59] in 1980 demonstrated that the actions of soluble and insoluble fibers on the colon depend on the extent to which they are digested [59]. Currently, dietary fiber consumption in the general population is still low, despite the guidelines’ recommendations [60].

Fibers are carbohydrates that derive from the plant cell wall and resist to gastric acidity, enzymatic hydrolysis and absorption in the upper gastrointestinal tract. Fibers act as nourishment for the gut microbiota, and some of them act as prebiotics, leading to the production of metabolites useful for human health. Accessible microbiota carbohydrates (MACs) resist digestion and are made available to the gut microbiota as prebiotics metabolizable into SCFAs. Generally, they are divided into soluble and insoluble, although there are some “fibrous foods” such as psyllium or oats which contain a certain amount of both fibers [61].

Fibers should be classified according to their fermentability, viscosity and gel-forming ability, for example:Insoluble and not very fermentable fibers (i.e., whole grains);Soluble, non-viscous and readily fermentable fibers (i.e., inulin);Soluble, gel-forming and non-fermentable fibers (i.e., psyllium) [61];

Insoluble fiber increases the fecal mass and colonic transit rate through mechanical stimulation of gut mucosa, inducing secretion and peristalsis [59] and having a significant laxative effect.

Soluble viscous fiber is minimally fermented and presents a high gel-forming capacity that is preserved throughout the large bowel, normalizing stool form especially in constipation. In fact, stool consistency is highly correlated to stool water content for its stool-softening effect [62].

In 2005, Rees et al., conducted a longitudinal, prospective, randomized, placebo-controlled trial, including 28 IBS-C patients that were equally randomized to a fiber intervention or a placebo group. The fiber intervention assumed 10–20 g/day of coarse wheat bran supplement to their normal diet vs. the low-fiber placebo for 8–12 weeks. In the end, the fiber intervention group had an increase in fecal wet weight compared to the placebo group. No significant differences for other intestinal function measurements and symptoms were observed between the two groups [63].

In 2011, Choi et al., conducted an RCT in IBS-C, IBS-D and IBS m patients. One hundred and forty-two patients were recruited and randomized to a fiber intervention group or to a placebo group for 4 weeks. The fiber intervention consisted in 150 mL of probiotic fermented milk with 3.15 g of fiber powder using sea tangle, radish and glasswort extracts, which are mostly soluble fibers, vs. 150 mL of probiotic fermented milk alone. Changes in Visual Analog Scale (VAS) scores were measured for abdominal pain or discomfort, abdominal distention or bloating, urgency, straining, a feeling of incomplete evacuation and improvement in overall IBS symptoms. The probiotic-fermented milk improved numerous parameters, and especially increased stool frequency in the IBS-C group [64].

The following year, Min et al., conducted II RCT in patients affecIed by IBS. One hundred and thirty patients were recruited, of which 65 were randomized to the fiber intervention or placebo for 8 weeks. The fiber intervention was composed of twice daily composite yoghurt with acacia dietary fiber, high-dose *B. lactis,* vs. a control product. Observing only the IBS-C subtype, 19 patients were in the treatment group, while 22 were in the control arm. The improvement in overall IBS symptoms was significantly higher in the test group than in the control one [65].

### 5.4. Functional Foods and Other New Approaches

There is a lack of data in the literature about the effects of functional foods, especially in the IBS-C subtype. Functional foods are “foods that offer health benefits extending beyond basic nutrition”. Among functional foods, some such as anthraquinones (present in senna, cascara, rhubarb and aloe vera), figs, kiwifruit, prunes linseeds have been studied for IBS, especially for the constipation subtype.

Anthraquinones are compounds derived from plants able to stimulate the motility and secretion of the colon; indeed, senna and cascara are particularly known for their laxative capacity. Among anthraquinones, there is also rhein, a compound contained in rhubarb. Even though rhubarb has been seen as a helpful supplement in improving stool frequency and consistency in constipation [66], in the literature, there are no RCTs in IBS-C [67].

Additionally, *aloe vera* has been studied and shown to be able to improve IBS symptoms (specifically in IBS-C patients), especially stool frequency and consistency [67].

Figs are usually suggested in cases of IBS-C, and indeed it has been demonstrated that they are useful for improving the microbiota composition, increasing the production of SCFAs, stool weight and consistency, and also for improving IBS-related symptoms [67]. In an RCT conducted by Pourmausomi et al., rehydrated figs (90 g/d) and flixweed (60 g/d) were compared with a placebo for 4 months in 150 IBS-C patients, and both interventions led to significant improvements in stool consistency and frequency, even if no results were observed regarding the severity of abdominal pain [68].

Additionally, kiwis are suggested in order to improve stool consistency, stool weight, the colonic microbiota and the production of short-chain fatty acids. This positive effect seems to be determined by the oxalate-soluble pectin present in kiwi fruit, which is fully absorbed in the small intestine, and the pectic fractions that are completely fermented in the colon. These characteristics of the different components of kiwifruit could be responsible for the increased water retention, fecal bulking and decreased transit time observed in pigs [69,70]. Moreover, it has been demonstrated in humans that consumption of freeze-dried green kiwi rapidly increased *Lactobacillus* and *Bifidobacterium* species, with a decrease in *Clostridium* and *Bacteroides* [71]. This effect, also demonstrated in an in vitro model of fermentation [72] and in animal models [73], is transient in nature [71]. This specific microbiota modulation could be able to reduce the presence of methanogenic bacteria, which seem to have a role in modifying colonic motility [47,48]. The limitation of these studies, also described by Bayer and colleagues [74], is the huge heterogeneity between the clinical protocols used, with different kinds of administration and the choice of the placebo or the control group. Therefore, further investigations are needed, even if the results are promising.

Dried plums or prunes are a well-known natural laxative. This laxative action may be due to their content of sorbitol, a sugar polyalcohol, which exerts an osmotic effect, and/or their fiber content that includes pectin, cellulose, hemicellulose, and lignin. Dried apricots also contain sorbitol and fiber, even if less than prunes [75].

Although there is not yet strong evidence regarding linseeds, and only a consensus agreement, they seem to improve digestive health or relieve constipation in IBS-C patients. Indeed, linseeds are rich in fiber and omega-3 fatty acids, as well as phytochemicals called lignans. Up to one [74] or two [49] tablespoon(s)/day of linseeds may be helpful for constipation, abdominal pain, and bloating. Like other sources of fiber, flaxseed should be taken with plenty of water or other fluids (150 mL fluid/tablespoon) [49].

## 6. Effect of Intestinal Microbiota Modulation in Patients with IBS-C

In recent years, various therapeutic strategies have been developed which are capable of modulating the gut microbiota composition, such as the use of prebiotics, probiotics or symbiotics, as well as the new frontier of fecal transplantation. As the modulation of intestinal microbiota seems to play a crucial role in several diseases, these possibilities are also explored in IBS-C patients.

### 6.1. Prebiotics in the Treatment of IBS-C

Prebiotics were firstly defined as “nondigestible food ingredients that beneficially affect the host by selectively stimulating the growth and/or activity of one or a limited number of bacteria in the colon, thus improving host health”. This definition was later refined to include other areas that may benefit from selective targeting of particular microorganisms: a selectively, generally, fermented ingredient that allows specific changes, both in the composition and/or activity in the gastrointestinal microflora, that confer benefits [76].

Although all prebiotics are fibers, not all fibers are prebiotics. Classification of a food ingredient as a prebiotic requires that the ingredient (1) resists gastric acidity, hydrolysis by enzymes, and absorption in the upper gastrointestinal tract; (2) is fermented by the gut microbiota, and (3) selectively stimulates the growth and/or activity of gut bacteria potentially associated with health and well-being [76].

As has already been stated, dietary fiber has soluble and insoluble fractions. Though the insoluble fiber is less utilized by the gut microbiota, the soluble ones, such as inulin and fructans, are mostly used by the gut microbiota as an energy source promoting the development of some beneficial bacteria such as *Lactobacillus* and *Bifidobacteria* [77]. In the literature, the enrichment of the genus *Prevotella* in individuals with higher fiber diets has also been shown [78].

*Prevotella* is an abundant genus in healthy people. Intestinal *Prevotella* spp. are commonly associated with diets and nutritional patterns rich in carbohydrates, resistant starch and fibers. In Western-style diets, Ruminococcaceae and Lachnospiraceae often degrade dietary fiber, although nutritional interventions with fiber-rich foods usually result in a *Prevotella* abundance increase. Indeed, having a *Prevotella*-rich gut microbiota improves weight loss, decreases cholesterol levels and limits the bifidogenic effect in individuals consuming a fiber-rich diet [79]. It is interesting to note that patients with IBS appeared colonized by different strains of *P. copri,* and a correlation between isolates and disease grading was observed [80].

The compounds identified as having the most prebiotic effects are the inulin-type fructans (FOS, inulin, oligofructose) and GOS, many of which are widely present in grains, vegetables and legumes [81].

GOS play a role in modulating immune function, and they also have anti-inflammatory effects. This could be linked also to the possible beneficial effects of these prebiotics on IBS patients, in which a microscopic inflammation of intestinal mucosa has been found. In general, it is well recognized that GOS have important effects on global IBS symptoms, but not on abdominal pain [82].

Prebiotics such as inulin or FOS are characterized as ‘functional fibers’ [83]. Inulin is a nondigestible oligosaccharide which behaves as a soluble fiber that is naturally found in more than thirty thousand of plants, including vegetables such as wheat, garlic, onion, chicory, artichoke, and asparagus. Thanks to its chemical configuration, inulin is resistant to hydrolysis by digestive enzymes, so it reaches the colon undigested and is further selectively fermented by colonic microbiota [84]. Inulin intake has been linked to the regulation of bowel peristalsis and transit, of stool consistency and frequency, as it produces changes in the composition and activity of the gut microbiota to the modulation of immune response, mineral absorption, satiety and bone weight [85].

Interestingly, a pilot study conducted in 2013 by Isakov et al. [57] on IBS-C patients who received inulin enriched-yogurt demonstrated an improvement in bowel habits and transit time in patients when compared with the consumption of a traditional yogurt [57]. Even the regular consumption of inulin-enriched fermented milk beverages showed a significant improvement in the consistency of stools in patients with IBS-C.

Moreover, Pilipenko et al., conducted an RCT on 49 patients which showed that the consumption of a functional drink containing 4 g of inulin, 4 mg of menthol, and 2 mg of pyridoxine is associated with improvements in stool parameters, abdominal pain, Bristol stool scale index and an increase in QoL in patients with IBS-C, but produces noticeable heartburn. A modification of the functional drink’s composition is necessary to reduce side effects [86].

Furthermore, Pilipenko et al., conducted another RCT on 50 patients fulfilling the Rome III criteria for IBS-C that were randomized into two groups: one received a standard diet plus two jelly drinks (containing 3 g of inulin, 10 mg of curcumin and 1.8 mg of pyridoxine) daily for 2 weeks, and a control group received only a standard diet. Likert scales were used daily to record abdominal pain, bloating, a feeling of incomplete bowel emptying, frequency of bowel movements and the Bristol stool scale. The jelly drinks’ consumption was associated with a significant beneficial effect on the stool parameters, a reduction in abdominal pain severity, bloating and in the sense of incomplete bowel emptying, as well as an increase in QoL. Patients in the control group showed improvements in abdominal pain and bloating only. During the treatment period, no significant adverse events were found [87].

### 6.2. Probiotics in Patients with IBS-C

Probiotics are defined by Dr. Roy Fuller as “live microbial feed supplements which beneficially affect the host, improving its intestinal microbial balance” [88]. Probiotics are live microorganisms that when administered in adequate amounts, confer a health benefit to the host [89].

Probiotic bacteria can replace a ‘missing part’ of the commensal microbiota, either in the small and/or large intestine, or stimulate a component of the existing commensal population [90]. Thanks to this action, the functionality of the microbiota might be restored, at least in part, leading to a symptom’s improvement. This might occur through several different pathways, such as competitive exclusion of other bacteria, the production of bacteriocins or an alteration in the fermentation capacity of the microbiota. Moreover, other studies in the literature have also demonstrated that probiotics may alter motility [91], reduce intestinal permeability [92], normalize the inflammatory profile (IL-10:IL-12) [93], reduce visceral hypersensitivity, attenuate anxiety behaviors and modulate brain activity in IBS subjects [94]. A recent systematic review and meta-analysis evaluating RCTs, conducted to assess the effects of probiotics in IBS patients, demonstrated a beneficial effect of these organisms in the treatment of this disorder [95].

Spiller et al., conducted a randomized, double blind, placebo-controlled study on 379 IBS subjects (IBS-C N = 180). Subjects were randomly supplemented with probiotics (*S. cerevisiae I-3856* at the dose of 1000 mg per day) or placebo for 12 weeks. *S. cerevisiae I-3856* did not improve intestinal pain and discomfort in IBS patients, except for the constipation subgroup; in fact, the number of complete spontaneous evacuations was higher in the intervention group, and the stools tended to be softer compared to placebo, suggesting that transit may have been accelerated. Moreover, also abdominal pain/discomfort and bloating improved in the IBS-C subtype throughout the study and at the end of the supplementation compared to placebo [96].

Mezzasalma et al., conducted a randomized, double-blind, three-arm parallel group trial on 150 IBS-C subjects divided into three groups (F_1, F_2, and F_3). This study aimed to evaluate the efficacy of two probiotic formulations on IBS-C symptoms [97]. Each group received a daily oral administration of probiotic mixtures for 60 days: F_1 (containing *L. acidophilus* and *L. reuteri*), F_2 (containing *L. plantarum, L. rhamnosus,* and *B. animalis* subsp. *lactis*) or placebo F_3, respectively. Fecal microbiological analyses were performed by species-specific qPCR to measure the amount of probiotics. The responders rate for each symptom was higher in the probiotic groups than to placebo both during the treatment and in the follow up (30 days after the end of the study). Probiotics increased during the times of treatment only in subjects treated with F_1 and F_2 but not with F_3, and the same level was maintained during the follow-up period In conclusion, the different species of probiotics administered to the IBS-C subjects constituted an important contribution to treating IBS-C symptoms [97].

Bahrudin et al., conducted an RCT to investigate whether the addition of polydextrose to sterilized probiotic containing *Lactobacillus helveticus* conferred benefits to IBS-C patients. A total of 163 patients were randomized in two groups: Group A consumed 350 mL of sterilized probiotic with 5.85 g polydextrose daily for 1 week, and Group B without polydextrose. The intestinal transit time, fecal pH, fecal weight, and pre- and post-consumption questionnaires were assessed. The addition of polydextrose to sterilized probiotic containing *L. helveticus* did not show significant benefits to IBS-C patients. However, the daily consumption of sterilized probiotic containing *L. helveticus* with or without polydextrose for a week alleviated constipation-related symptoms and reduced both fecal pH and intestinal transit time [98]. In an interesting randomized cross-over case–control study, Bărboi et al., included 51 IBS-C patients, of which 47 completed the trial. Patients were randomized into two groups receiving a diet specific for constipation with or without a food supplement containing inulin, choline and silymarin. Patients were evaluated at baseline, after 4 and 8 weeks, using a questionnaire to assess IBS symptoms. In the supplemented group, abdominal pain and abdominal bloating severity improved by 68.3% and 34.8%, respectively. Even if both the evacuation frequency per week and the stool consistency according to the BSS improved in both groups, no significant differences were observed between the two groups. In conclusion, the combination of inulin, choline and silymarin associated with a diet specific for constipation showed clinical beneficial effects on IBS-C patients in terms of bowel movement, abdominal pain and bloating [99].

### 6.3. Symbiotic in Patients with IBS-C

In 1995, Gibson and Roberfroid introduced the term “symbiotic” to describe a combination probiotics and prebiotics that act synergistically. A symbiotic should exert a synergistic benefit, enhancing the probiotic organisms by the selective, co-administered prebiotic substrate. Therefore, a correct combination of both components in a single product should ensure a superior effect compared to the activity of the probiotic or prebiotic alone [76]. In 2013, Cappello et al., conducted a double-blinded, randomized placebo-controlled study to evaluate the effects of a commercially available multi-strain symbiotic mixture (Probinul, 5 g over 4 weeks) on symptoms, colonic transit and QoL in IBS patients who met the Rome IV criteria [100]. A total of 64 patients were randomized to either placebo (*n*  =  32) or symbiotic (*n*  =  32), and the symbiotic mixture contained lyophilized bacteria (5 × 10^9^ *Lactobacillus plantarum*, 2 × 10^9^ *Lactobacillus casei* subp. *rhamnosus* and 2 × 10^9^ *Lactobacillus gasseri*, 1 × 10^9^ *Bifidobacterium infantis* and 1 × 10^9^ *Bifidobacterium longum*, 1 × 10^9^ *Lactobacillus acidophilus*, 1 × 10^9^ *Lactobacillus salivarus* and 1 × 10^9^ *Lactobacillus sporogenes* and 5 × 10^9^ *Streptococcus termophilus*), prebiotic inulin (2.2 g) and 1.3 g of tapioca-resistant starch. The study preparation was administered in a powder form (5 g sachets) containing the symbiotic mixture or the matching placebo. The two sachets were comparable in color, texture and taste. The patients were instructed to ingest the preparation twice daily, far from meals, dissolved in water. Global satisfactory relief of abdominal flatulence and bloating were the primary endpoints, while changes in abdominal bloating, flatulence, pain and urgency, stool frequency and bowel functions on BSS and sense of incomplete evacuation were the secondary endpoints. Additionally, pre- and post-treatment colonic transit time and QoL were evaluated. After 4 weeks, the symbiotic group showed a reduced flatulence, a longer rectosigmoid transit time and an improved QoL [100]. In conclusion, the symbiotic mixture failed to satisfy the primary endpoints, but it demonstrated a beneficial effect on flatulence in IBS patients. The mixture, however, showed a lack of any adverse events and a good side-effect profile. In the current literature, few studies have evaluated the relationship among symbiotics, microbiota and IBS symptoms, but without specifying the IBS subtype. Further studies on a larger number of patients are needed to confirm whether a symbiotic mixture might be an effective treatment option in IBS.

### 6.4. Fecal Microbiota Transplantation in Patients with IBS-C

Fecal microbiota transplantation (FMT), also known as fecal bacteriotherapy or fecal infusion, consists of administration of a liquid filtrate of feces from a healthy donor into the GI tract of a recipient person. Increasing evidence supports the role of the gut microbiota in the etiology of irritable bowel syndrome (IBS). Fecal microbiota transplantation seems to be a highly effective treatment against the recurrent infection of *Clostridioides difficile,* as shown in RCTs in the literature, and may be beneficial also in case of ulcerative colitis. However, its efficacy in IBS is not well defined. In the single-center, retrospective study conducted by Cui et al., in 2021, the long-term efficacy of fecal microbiota transplantation (FMT) in patients with moderate to severe IBS was investigated [101]. They evaluated treatment efficacy rates, changes in IBS-SSS, IBS-specific quality of life and fatigue, effect on stool frequency, Bristol Stool Scale for IBS-C and IBS-D, and the side effects. Overall, 100 g of stool suspension was administered through a naso-intestinal tube or colonoscopy within 6 min/daily for six consecutive days. The stool frequency of IBS-C patients increased from 1.5 ± 1.38 times per week to 2.68 ± 1.15 times per week one month after FMT treatment and increased to 4.33 ± 1.56 times per week at the end of the fifth year of follow-up. The BSS score of IBS-C patients significantly increased (*p* < 0.05) from 2.13 ± 0.88 before treatment to 2.94 ± 1.3 one month after FMT treatment, and further increased to 3.71 ± 1.21 by the 5th year after FMT (compared with that before FMT, *p* < 0.01) [101].

Fecal microbiota transplantation (FMT) seems to be a promising treatment for IBS patients. Although in Western countries, females present a higher prevalence of IBS, El-Salhy et al. did not find a sex difference in the response to FMT either in the placebo group or the actively treated group in their study in 2021. They included 164 IBS patients with moderate-to-severe IBS symptoms belonging to the IBS-D, IBS-C and IBS m subtypes, and who had not showed improvement in symptoms after the NICE-modified diet. Patients were divided into three groups: the placebo (own feces) and two actively treated groups (30 g or 60 g superdonor feces). The results appear to be more effective in IBS-D patients compared to IBS-C ones [102].

## 7. Conclusions

In conclusion, the relationship between IBS symptoms and food ingestion is astoundingly complicated. Currently, there is not a specific diet to manage IBS-C [103]; nevertheless, the nutritional approach represents the first line of intervention to improve IBS symptoms [49].

To date, the low-FODMAP diet is the most widely used diet to manage IBS. However, the beneficial effect on symptoms appears to be higher in IBS-D than in IBS-C patients. Although the FODMAP-restricted diet may be effective in the short-term management of some IBS-C patients, more trials are needed to establish also its long-term efficacy and safety, investigating especially colonic health and microbiota [104].

Precisely because to date there is not an “ideal diet” for IBS subjects, clinical practitioners usually suggest general nutritional advice. Moreover, among the various nutritional alternatives, the intake of probiotics [90] is suggested. The intake of probiotics showed a positive effect both on the consistency of the stool and on the intestinal transit, improving the QoL of IBS-C patients [105].

Although eating habits’ modification does not improve abdominal pain, which is the main recognized symptom associated with IBS, what has emerged overall from the studies in the literature is that a healthy lifestyle and a balanced diet certainly lead to an improvement in abdominal pain, which results in a better quality of life for patients.

Further studies are needed to figure out the most appropriate nutritional protocol for reducing IBS-C symptoms through improving the microbiota composition.

## Figures and Tables

**Table 1 nutrients-15-01647-t001:** Microbiota composition in IBS-C subjects versus healthy controls (HC).

References	Genera	IBS-C Patients vs. Healthy Controls
Malinenet al. [40]	*Veillonella* spp.	IBS-C: ↑
Maukonen et al. [41]	*Clostridium coccoides*-*E. rectale* group	IBS-C: ↓
Rajilić-Stojanović et al. [42]	Firmicutes (*Clostridium*) Bacteroidetes Actinobacteria	IBS-C: ↑ IBS-C: ↓ IBS-C: ↓
Chassard et al. [43]	Enterobacteriaceae Sulfate-reducing bacteria *Bifidobacterium* *Lactobacillus*	IBS-C: ↑ IBS-C: ↑ IBS-C: ↓ IBS-C: ↓
Durbán et al. [44]	Bacteroidetes Enterobacteriaceae	IBS-C: ↑ IBS-C: ↑
Parkes et al. [45]	Bacteroidetes *Bifidobacterium* *C.coccoides*-*Eubacterium rectale*	IBS-C: ↑ IBS-C: ↑ IBS-C: ↑

Legend: ↑ increase; ↓ decrease.

## Data Availability

Not applicable.

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
