# Peer review of "Constipation-Predominant Irritable Bowel Syndrome (IBS-C): Effects of Different Nutritional Patterns on Intestinal Dysbiosis and Symptoms"

_nutrients, 2023, doi:10.3390/nu15071647_

Round 1
Reviewer 1 Report
Drs Di Rosa and team provide a narrative review on IBS with predominant constipation and the effects of intestinal microbiota and dietary interventions on symptoms.
The article is well-structured. It is comprehensive and includes all relevant aspects of intervention in this field of constipation-predominant IBS.
Some aspects that can be improved:
-section on `Human gut microbiota` is quite extensive and long and could be shortened.
In general:
-vocabulary and style: Some expressions are difficult to understand (LL162-it is not clear what the authors mean here)
L210: suggest: ...directly causes improvement of symptoms...
L227: general advice (not advices)[see also L301] ...advice was in accordance...
L230: ..should be warned or instructed about ....
Author Response
Dear Reviewer 1,
thank you for your comments. Please, attached you can find the corrections.

Reviewer 2 Report
Line 91 and diversity increase gradually FROM the stomach to the colon (1011 cells/ml) [17]. The micro-
Ref 32: should be updated with this: https://pubmed.ncbi.nlm.nih.gov/32835795/
Individual characteristics such as food intolerances should be mentioned, as a cause for IBS symptoms.
In fact, methane decreases ileal and 208 colonic transit time and increases the amplitude of contraction, slowing peristalsis and 209 causing constipation. EXPLAIN HOW A DECREASED TRANSIT TIME INCREASES CONSTIPATION - IT DOESN’T MAKE SENSE LOGICALLY.
Author Response
Dear Reviewer 2,
thank you for your comments. Please, attached you can find the corrections.

Reviewer 3 Report
The objective of this review was to analyze the role of the diet of certain nutritional components, such as prebiotics, probiotics and symbiotics, among others, on the symptoms and microbiota of individuals with irritable bowel syndrome of the predominant constipation subtype. Overall, the material is good, the sections are adequate and the language is clear. However, the manuscript needs some adjustments before it can be accepted for publication. The authors did not disclose a limitation of the time period of the references; which keywords were used in the survey and which databases were analyzed. Also, there are some spelling and italic errors, of which I mention some that I found, but authors should check the entire manuscript. Here are some comments:
Minor comments:
Line 22: please, separate “clearbut”;
Line 88: “houses” is missed;
Line 115 (see also lines 194, 198, 414): “bifidobacteria” should not be in italics. Authors may also use the scientific designation of the genus, but in italics (Bifidobacterium);
Line 185 (see also lines 194, table 1): of “lactobacilli” should not be italicized. Authors may also use the scientific designation of the genus, but in italics (Lactobacillus);
Line 193 (see also lines 196, 213): Scientific writing of family names should not be italicized;
Line 294: “worsening” is missed;
Line: 299: “hundred” is missed;
Line 313: Psyllium should be italicized;
Line 398: Aloe vera is a species name and should be italicized
Line 415: Please use “Clostridia” or “Clostridium”. The latter in italics;
Line 461: Family names should not be italicized;
Line 477: “resistant” is missed;
Lines 514-515: Authors should please use only "bacteriocins", as this expression already indicates compounds with antimicrobial activity;
Line 538: Separate “than to”;
Line 562: Separate “showed clinical”;
Line 586: Separate “a reduced”;
Line 611: Remove parenthesis;
Author Response
Dear Reviewer 3,
thank you for your comments. Please, attached you can find the corrections.
